# A Combination of Pharmacophore-Based Virtual Screening, Structure-Based Lead Optimization, and DFT Study for the Identification of *S. epidermidis* TcaR Inhibitors

**DOI:** 10.3390/ph15050635

**Published:** 2022-05-21

**Authors:** Srimai Vuppala, Jaeyoung Kim, Bo-Sun Joo, Ji-Myung Choi, Joonkyung Jang

**Affiliations:** 1Department of Nanoenergy Engineering, Pusan National University, Busan 46241, Korea; srimaiv@pusan.ac.kr (S.V.); k960628@pusan.ac.kr (J.K.); 2Infertility Institute, Pohang Women’s Hospital, Pohang 37754, Korea; 3Lab-to-Medi CRO, 12 Dosandae-ro-8-gil, Seoul 06038, Korea; kjcjm@hotmail.com

**Keywords:** TcaR, biofilm, drug design, bacterial infections, molecular docking

## Abstract

The transcriptional regulator (TcaR) enzyme plays an important role in biofilm formation. Prevention of TcaR-DNA complex formation leads to inhibit the biofilm formation is likely to reveal therapeutic ways for the treatment of bacterial infections. To identify the novel ligands for TcaR and to provide a new idea for drug design, two efficient drug design methods, such as pharmacophore modeling and structure-based drug design, were used for virtual screening of database and lead optimization, respectively. Gemifloxacin (FDA-approved drug) was considered to generate the pharmacophore model for virtual screening of the ZINC database, and five hits, namely ZINC77906236, ZINC09550296, ZINC77906466, ZINC09751390, and ZINC01269201, were identified as novel inhibitors of TcaR with better binding energies. Using structure-based drug design, a set of 7a–7p inhibitors of *S. epidermidis* were considered, and Mol34 was identified with good binding energy and high fitness score with improved pharmacological properties. The active site residues ARG110, ASN20, HIS42, ASN45, ALA38, VAL63, VAL68, ALA24, VAL43, ILE57, and ARG71 are playing a promising role in inhibition process. In addition, we performed DFT simulations of final hits to understand the electronic properties and their significant role in driving the inhibitor to adopt apposite bioactive conformations in the active site. Conclusively, the newly identified and designed hits from both the methods are promising inhibitors of TcaR, which can hinder biofilm formation.

## 1. Introduction

Nowadays, bacterial infections pose a major threat to society. In particular, Staphylococci are the most frequent cause of bacterial infections in humans worldwide [1]. The *Staphylococcus aureus* and *Staphylococcus epidermidis* often cause infections in immunocompromised people because these bacteria are mainly situated on the skin and mucous surfaces of humans [1,2]. Especially *S. epidermidis* is the most frequently isolated microorganism from human epithelia and is also known as a Gram-positive and coagulase-negative staphylococcus [1,3]. It is considered an important opportunistic pathogen, which is an accountable reason for nosocomial infections or health-care-associated infections due to extensive usage of medical implants and devices. *S. epidermidis* is the skin colonizer of humans and other mammals; due to that reason, there is a high probability of medical device contamination during the insertion process [1,2,3]. Both the bacteria *S. aureus* and *S. epidermidis* can form a biofilm (accumulation of one or more microorganisms that can grow on different surfaces), which is a pressing issue in the development of medical device-related infections and health-care-associated infections in hospital patients [4,5,6,7]. These infections are a serious clinical issue, especially given that a large percentage of hospital patients are subjected to procedures for inserting foreign devices, such as pipes and artificial heart valves. The treatment for nosocomial infections associated with biofilm formation is extremely difficult, especially in the case of urinary tract infections where more than 40% of infections are related to the biofilm-forming bacteria [2,5,8]. Biofilm formation in wounds can extend the healing time and also lead to further infection. In the case of implants, the biofilm can change the effectiveness of the medical device and lead to the removal or replacement of the implant. The National Institute of Health reported that nearly 80% of microbial infections are caused by biofilms [5,9]. Polysaccharide intercellular adhesion (PIA) is a crucial component in *S. epidermidis* of biofilm extracellular matrix, which is composed of a homopolymer of β-1, 6-linked N-acetylglucosamine [10]. Recent studies reported that the production of PIA is governed by the expression of the *icaADBC* operon, which is regulated by transcription regulator TcaR [11,12,13]. TcaR belongs to the MarR family of genes that is involved in resistance to teicoplanin and methicillin and is also a multifunctional regulator, which is not only associated with *icaADBC* transcription [14]. Recent research by Yu-Ming et al. on TcaR with several antibiotics reported that the antibiotics interacted with TcaR at the binding site of the winged helix-turn-helix motifs (DNA binding domain) and weaken the binding affinity to its target DNA finally promoting the exit of TcaR from its *ica* promoter. They confirmed from the results of TcaR-antibiotic complexes, that antibiotic treatment inhibits the formation of the TcaR-DNA complex in *ica* operon. In addition, they have revealed that the allosteric mechanism is responsible for inhibiting the production of PIA, which is an essential constituent of biofilm formation [15]. Design and discovering the novel ligands that bind more strongly and specifically to the TcaR in the *S. epidermidis*, restricting their DNA binding would shut down the production of PIA and the formation of biofilm (Figure 1). Antibiotic resistance capability to biofilm is also the most important and rising problem throughout the world. To address this problem, it is an urgent need to design and develop novel pharmacophores with good binding affinity and high structural diversity compared to known antibiotic drugs.

Drug design and discovery is a laborious, expensive, and time-consuming process. Therefore, the application of computer-aided drug design techniques, such as ligand-based and structure-based techniques, are of huge importance in decreasing the cost and time as well as intensifying the efficiency of drug discovery research [16]. Herein, we applied two efficient strategies that can incorporate the advantage of pharmacophore modeling (ligand-based) and structure-based lead optimization based on molecular docking simulations to identify the ligands that contain important chemical features and strong binding affinity for *S. epidermidis* TcaR. A pharmacophore modeling-based virtual screening technique was successfully applied to an FDA-approved drug (gemifloxacin) to identify the best binding affinity ligands from the ZINC 15 chemical database, which contains 22 million compounds with 200 million biologically active conformations, against *S. epidermidis*. To employ the second strategy, which is structure-based lead optimization, we took a set of experimentally known inhibitors of *S. epidermidis,* and we also validated our in silico methodology and results with experimental inhibitory activity of selected dataset. Therefore, our study can assist the experimentalists in their in vitro and in vivo analysis. Finally, by applying a series of computational simulations, six drug-like compounds (five and one from the first and the second strategy, respectively) are identified as possible strong binding ligands of *S. epidermidis* TcaR. Further, to understand molecular interactions, we performed the density functional theory (DFT) simulations on the final six-hit compounds to calculate their electronic properties, such as HOMO, LUMO, and molecular electrostatic potentials. Finally, the outcome of our research work establishes how pharmacophore modeling and structure-based lead optimization accompanied with molecular docking simulations can be an efficient methodology to identify the novel hit compounds with high structural diversity that can bind in the active site of the target.

## 2. Results

### 2.1. Ligand-Based Pharmacophore Modeling

#### 2.1.1. Generation of Pharmacophore Model

To identify the hit compounds as potent drug candidates for TcaR, the pharmacophore modeling strategy was applied to gemifloxacin, which is the most potent drug for *S. epidermidis* bacterial infections. Here, we considered two pharmacophore models of gemifloxacin. The first model is generated by using the geometry optimized conformation of gemifloxacin, whereas the best (lowest binding energy) conformation of gemifloxacin against *S. epidermidis* TCAR was used to generate the second model. Figure 2 represents the two pharmacophore models of gemifloxacin. The first model was generated by five pharmacophore features, i.e., hydrogen bond acceptor, negative ion charge, and three hydrophobic regions, whereas the second pharmacophore model was characterized by six features, i.e., one negative ion charge and five hydrophobic regions.

These two pharmacophore models were generated by an open-source tool ZINCPharmer to rapidly screen through the ZINC 15 database. This resulted in 708 hits being identified out of more than 22 million compounds with 200 million conformations. After that, the screened 708 compounds were filtered by applying filtering parameters (molecular weight less than 500, maximum hits per conformation of each molecule 1, and the number of rotatable bonds < 15). We identified 308 hit compounds that are used for further screening through molecular docking simulations by using AutoDock tools. The 308 hits obtained using pharmacophore modeling were retrieved from the ZINC 15 database and saved as a single SDF file format and then converted to the PDBQT format by using the open-source tool Open Babel.

#### 2.1.2. Molecular Docking Simulations for the Identification of Hit Compounds

Validation of Ligand Binding Mode

The 308 hit compounds identified from virtual screening of the ZINC 15 database were further filtered using AutoDock tools by evaluating all the molecular interactions between identified compounds against *S. epidermidis* TcaR. Using AutoDock, Gasteiger charges as well as the maximum number of torsions of 5 were added to the ligands. The docking active site is defined through a grid dimensions coordinates: center_x = −22.324 Å, center_y = −29.265 Å, and center_z = −0.294 Å. We constructed a grid box by pointing 62, 64, and 74 points in the x, y, and z directions, respectively, with a grid point spacing of 0.292 Å. The docking analysis was performed with Lamarckian genetic algorithm with a number of genetic algorithm runs set at 10 and the other docking parameters set to default values. Using AutoDock scoring functions, ten different conformations of every ligand were generated and ranked according to their binding energies. The estimated inhibition constant (Ki) was calculated to estimate which compounds are able to inhibit TcaR at lower experimental concentrations. The Ki is calculated from the binding energy (ΔG) by applying the formula Ki = exp(ΔG/RT), where R is the universal gas constant (1.985 × 10^−3^ kcal mol^−1^ K^−1^), and T is the temperature (298.15 K). To confirm the grid dimensions and validate the docking accuracy, we redocked the methicillin, a known and crystallographically observed inhibitor of *S. epidermidis* TcaR and reproduced the superimposed docking image of *S. epidermidis* TcaR-methicillin complex with the crystal structure of *S. epidermidis* TcaR-methicillin (PDB ID: 3KP4) (Appendix A).

Identification of Hit Compounds

After the validation of ligand (methicillin) binding mode and docking protocol, the molecular docking simulations were performed for 308 compounds against *S. epidermidis* TcaR. As shown in Table 1, we selected the best 16 hits that showed better binding energies (less than −10.6 kcal/mol) compared to that of gemifloxacin, which were then considered for the further screening process. Of these 16 compounds, one FDA-approved drug ZINC01269201 (Prulifloxacin), which can be used to treat some bacterial infections, and also six novel compounds, namely ZINC77906466, ZINC77906236, ZINC72332562, ZINC21985520, ZINC03114214, and ZINC01958447 (not fluoroquinolones), with good binding energies and estimated inhibition constant (Ki) against *S. epidermidis* TcaR were identified. Two hits, namely, ZINC09550296 and ZINC01270492 were identified commonly in two pharmacophore models.

To identify the best hits, we performed further screening analysis by calculating the physicochemical parameters, drug-likeness, and ADMET properties. Appendix A report that except for three hits (ZINC03114214, ZINC02280291, and ZINC01958447) all other compounds are well-qualified by Lipinski’s rule of five and ADMET properties. Finally, from a series of computational simulations, we identified a total of five compounds, which include ZINC77906466 and ZINC77906236 (non-fluoroquinolones), ZINC09751390, ZINC09550296, and ZINC01269201 (prulifloxacin). They well-satisfied the Lipinski’s rule of five with good drug-like (DL) scores and showed better binding energies than that of gemifloxacin.

Binding Mode Analysis of Final Hits

Molecular docking results reveal the binding mode and interaction mechanism of final hits and gemifloxacin in the active site of TcaR. The gemifloxacin bound in the active site in a different mode compared to that of the known crystallographic inhibitor drug methicillin (binding energy of −6.25 kcal/mol, Ki of 26.35 uM) attained strong binding affinity. As shown in Figure 3, methicillin formed two H-bond interactions and one π-alkyl interaction with GLN61, GLN31, and HIS42, respectively. Detailed investigation of the molecular interactions revealed that gemifloxacin (Figure 4) formed four H-bonds and one π-cation interaction with ASN45 (two H-bonds), ASN20, GLU39, and ARG110, respectively. The quinolone ring of gemifloxacin formed a salt bridge with HIS42 and acquired binding energy of −10.73 kcal/mol, with an estimated inhibition constant of 13.72 nM.

The superimposition of the final five compounds, gemifloxacin, and methicillin inside the TcaR active site revealed that the four compounds followed the binding mode of gemifloxacin, and one compound, namely ZINC77906236, occupies the binding mode of both drugs (Figure 5a–c).

Binding Mode Analysis of Hit Compound ZINC77906236

Out of the five final hits, the compound ZINC77906236 has shown better binding energy of −13.27 kcal/mol and Ki of 187.61 pM. As shown in Figure 6, this compound formed two H-bond interactions with ARG110 and another H-bond interaction with ASN45, whereas four other interactions, such as π-π T-shaped, two π-cation, and π-donor H-bond, were observed with the key interacting active site residue HIS42. The presence of an important electrostatic π-cation and π-donor H-bond interactions lead to the binding orientation of the ZINC77906236 compound in a more advantageous direction, which initiated the interactions with other active site residues, such as ALA24 and ALA38. This compound is also involved in five hydrophobic interactions (alkyl and π-alkyl) with ALA24, ALA38, and ILE16. This compound was identified by the first pharmacophore model from the database, and Appendix A represents the overlay of pharmacophoric features of the first model with hit compound ZINC77906236. It is important to note that this compound does not belong to fluoroquinolones; nowadays, fluoroquinolone antibacterial drug resistance has become a major issue [17,18]. To overcome this problem, identification of novel inhibitors as antibacterial drugs has become a state-of-the-art work. Thus, we identified a novel hit compound ZINC77906236 with better binding energy, Ki, and well-satisfied pharmacological properties.

Binding Mode Analysis of Hit Compound ZINC09550296

The compound ZINC09550296 is considered the second hit in the final five hits with the binding energy of −12.89 kcal/mol and Ki of 353.69 pM. As shown in Figure 7, this compound has established a greater number of close contacts with the active site residues of TcaR that lead to the four crucial H-bond interactions with HIS42, ASN45, ASN20, and ARG110 and hydrophobic interactions (π-π stacked, alkyl, π-alkyl) with ALA24, ALA38, HIS42, VAL63, and VAL68. The fluorine atom of this compound also formed a halogen bond interaction with the key interacting active site residue ASN20 that can enhance the binding efficacy and affinity [19] of the hit compound against TcaR. Additionally, this compound is also involved in two carbon-hydrogen bond interactions with active site residues ALA38 and GLU39. This hit was identified by the second pharmacophore model from the database and satisfied the all pharmacophoric features of the second model (Appendix A).

Appendix A represents the binding orientation of the other three final hits ZINC77906466 (non-fluoroquinolone), ZINC09751390, and ZINC01269201 (Prulifloxacin) in the active site of TcaR, and their corresponding binding energies and Ki are given in Table 1.

### 2.2. Structure-Based Lead Optimization Studies

#### 2.2.1. SAR and ADMET Analysis of Selected Experimentally Known Inhibitors

We performed the in silico analysis of 14 (7a–7p; Appendix A) experimentally known fluoroquinolone inhibitors of *S. epidermidis* to understand their structural features required to interact with the selected target by using ADMET and binding affinity prediction tools. We calculated the physicochemical and ADMET properties of the selected dataset by using Molinspiration and Osiris property explorer. All molecules satisfied Lipinski’s rule of five and showed a positive enzyme inhibitor constant, but the DL scores were very poor (Appendix A). The poor DL scores represented that these molecules are highly toxic and not good pharmacophores. We therefore moved to design novel pharmacophore analogs with improved pharmacological properties.

#### 2.2.2. Molecular Docking Analysis of Selected Experimentally Known Inhibitors

We have performed molecular docking simulations on *S. epidermidis* TcaR to understand the enzyme-ligand interaction at the molecular level and to find a suitable orientation for each ligand within the active site. The fitness scores obtained from the GOLD program were high for active molecules when compared to those of least active and inactive molecules. In the docking results of the selected dataset (7a–7p), the fitness scores and binding energies did not correlate with the inhibitory activity of the molecules, whereas the hydrophilic character (H-bond score) of molecules played an essential role and also exhibited a good correlation with their inhibitory activities. The most active molecules, namely 7a, 7b, and 7g, showed the highest protein-ligand H-bonding scores of 6.27, 6.73, and 6.40; fitness scores of 56.33, 57.47, and 56.90; and binding energies of −8.7, −9.0, and −9.2 kcal/mol (Table 2), respectively. The inactive molecules had low protein-ligand H-bonding scores except for molecule 7o. Accordingly, a significant correlation has been found between the protein-ligand H-bond score and the inhibitory activity for the selected dataset.

The binding mode and molecular interactions of the active molecules 7a, 7b, and 7g with the target protein are shown in Figure 8, Figure 9 and Figure 10, respectively. The two carboxyl groups of molecule 7a had five H-bond interactions with ARG110 (H-bond length (BL) 2.61 Å, 2.60 Å), GLU13 (BL 2.28 Å), and ASN20 (BL 2.44 Å, 2.55 Å). In molecule 7a, the π cloud of the quinolone ring is involved in two π-π stacked interactions (BL 3.74 Å, 3.97 Å) and two π-cation interactions (BL 4.10 Å, 3.74 Å) with HIS42 residue, and the Pyrone ring showed one π-Alkyl interaction with ALA38 (BL 4.41 Å), and one carbon-hydrogen bond interaction with GLN61 (BL 2.88 Å). Similarly, molecule 7b had five H-bond interactions with ARG110 (BL 2.63 Å), GLU13 (BL 2.84 Å), ASN17 (BL 2.97 Å), and ASN20 (BL 2.54 Å, 2.57 Å). Two carboxylic acid groups and one keto group were involved in these H-bond interactions. The π cloud of the quinolone ring has shown two π-π stacked interactions (BL 3.86 Å, 4.00 Å) and two π -cation interactions (BL 3.82 Å, 4.28 Å) with HIS42 residue, and the substituent pyrrole ring was involved in π-Alkyl interactions with VAL63 (BL 5.45 Å), and ALA38 (BL 4.17 Å). The keto group, two carboxyl groups, and six-membered piperazine ring of molecule 7g showed eight H-bond interactions with ARG110 (BL 2.61 Å, 2.33 Å), ASN45 (BL 2.62 Å), GLU13 (BL 2.39 Å), ASN17 (BL 2.61 Å), ASN20 (BL 2.49 Å, 2.72 Å), and GLU61 (BL 2.06 Å). The π cloud of the quinolone ring was involved in two π-π stacked interactions (BL 3.88 Å, 3.95 Å) and one π-cation interaction (BL 3.66 Å) with HIS42 residue. The present molecular docking analysis helped us to understand how each substituent affects the binding affinity with the target.

#### 2.2.3. Lead Optimization Studies

The initial pharmacological analysis of the selected dataset has shown poor DL properties. Therefore, we have carried out lead optimization studies by considering the most active inhibitors (7a, 7b, and 7g) as leads to develop novel molecules with improved pharmacological properties. We designed forty new pharmacophore analogs (Appendix A) by substituting various functional groups at different positions (1st, 6th, and 7th) of the basic skeleton of quinolone. On the basis of the structural–activity relationship, the antibacterial activity of quinolones significantly improved by the modification of different functional groups at the 1st, 6th, and 7th positions of basic skeleton [20,21,22]. Furthermore, we analyzed the importance of each substituent and how the substituent enhances their medicinal values in the basic skeleton by using the in silico tools.

SAR and ADMET Analysis of Designed Molecules

The physicochemical properties and in silico drug-relevant properties of designed molecules are summarized in Appendix A. The designed molecules satisfied Lipinski’s rule of five, a rule of thumb to evaluate the drug-likeness of a molecule. The lipophilicity values of the designed molecules were less than five. The number of the H-bond donors and acceptors was not more than five and ten, respectively. The molecular weight was less than five hundred Dalton. The designed molecules showed a positive enzyme inhibition constant, signifying that the molecules act as enzyme inhibitors.

The molecules of our design showed significantly higher DL scores (except Mol37 to Mol40) over the selected dataset. The positive DL scores (4.83 to 7.20) of these molecules confirm that these pharmacophore analogs qualify as potential commercial drugs. Molecules Mol36, Mol35, Mol32, Mol24, and Mol33 showed the highest DL scores of 7.20, 6.86, 6.37, 6.34, and 6.33, respectively. Interestingly, the in silico ADMET predictions indicated that the loss of one –COOH group (at 1st position) and the substitution of fluorine with chlorine can increase the DL properties and reduce the toxicity risks comparatively.

Molecular Docking Analysis of Designed Molecules

As per the docking results, all the designed molecules showed good fitness scores (<50) (Appendix A) against the target. In the selected dataset, the H-bond score showed a very good correlation with inhibitory activity. Therefore, molecules Mol1, Mol2, Mol4, Mol9, Mol17, Mol24, Mol34, and Mol35 with the highest H-bond scores were taken to be the most active inhibitors of the target.

We have drawn in Figure 11 the binding conformation of the best candidate molecule Mol34 in the active site of *S. epidermis* TcaR. The molecule (binding energy of −10.6 kcal/mol, H-bond score of 7.98, and fitness score of 62.95) took a conformation that fits well in the entire groove of the binding site of *S. epidermis* TcaR. The carboxyl and keto groups of Mol34 played a significant role in the binding by forming four H-bond interactions with the active site residues. The carboxyl group had two H-bond interactions with the ARG110 (BL 2.79 Å, 2.69 Å), and the keto group had one H-bond interaction with ASN20 (BL 3.00 Å). We observed that quinolone moiety was involved in two π-π stacked interactions, one H-bond interaction, and a π-cation interaction with HIS42 (BL 3.90 Å, 4.18 Å, 3.09 Å, and 4.0 Å respectively), and other substituted groups (benzyl ring and -C(CH_3_)_2_) are involved in several hydrophobic π-alkyl interactions with VAL63 (BL 4.85 Å), VAL68 (BL 4.89 Å), ARG71 (BL 5.20 Å), ALA24 (BL 4.31 Å), and ILE57 (BL 5.48 Å). Hit compounds identified from pharmacophore modeling-based virtual screening and structure-based lead optimization methods and their intermolecular interactions within the active site are shown in Table 3.

### 2.3. Density Functional Theory Calculations

We performed DFT simulations to understand deeper interactions of the final six hit molecules at the molecular level. The six molecules were optimized at the B3LYP/6-31G(d) level of theory [23,24] using Gaussian 16 quantum chemical simulation program [25]. The minima on the potential energy surface were further confirmed by their frequency analysis with no imaginary frequencies. The isodensity plots of frontier molecular orbitals and molecular electrostatic potential surfaces were generated from a single-point population analysis. The isodensity surfaces facilitated us to distinguish the interacting reactive sites of the molecules. The three-dimensional isodensity surfaces and MEP plots of the hit compounds are shown in Figure 6, Figure 7, Figure 11 and Appendix A. The hit compounds have not shown a similar electron density distribution pattern because of their structural diversity. From the isodensity plots, the electron density in HOMO is localized over or near the electron donor parts, such as alky-substituted N in the case of ZINC09550296. On the other hand, the electron density in LUMO is shifted towards the electron-withdrawing groups, such as the carboxylic group in the case of Mol34 and ZINC09751390. It is interesting to note that the moieties where the LUMO is localized are having the maximum number of interactions with the active site residues.

The MEPs of ligands play a very important role in binding and interaction with *S. epidermidis* and significantly affect the inhibition. Calculating the MEPs is considered the best way to identify the negative, neutral, and positive electrostatic potential areas in the molecule, and the areas are displayed with a unique color code. In the MEP surfaces, the utmost negative, neutral, and utmost positive electrostatic potential areas are represented by the red, green, and blue colors, respectively. We observed the maximum negative electrostatic potentials (MEP_min_) around the electronegative oxygen atoms of the carboxyl/carbonyl groups and the maximum positive electrostatic potentials (MEP_max_) around the nitrogen or alkyl-substituted nitrogen atoms. Interestingly, the aromatic five-membered and six-membered rings of the hit compounds neither fall in the most electronegative region nor fall in the most electropositive region; rather, they displayed green color on MEP surfaces. The H-bond interactions of the hit molecules with the active site are mainly found in the areas of maximum negative electrostatic potentials. From the MEPs of hit compounds, the electronegative area can act as an electron donor, whereas the electropositive area can act as an electron acceptor to the active site of the *S. epidermidis*. Our molecular docking results also affirmed the involvement of these areas in various interactions with the active site residues.

## 3. Discussion

Staphylococci are the most common bacterial infections in humans around the globe, in particular *S. epidermidis*, as it frequently causes infection in the immunocompromised people or after trauma to the epithelium and is more capable of producing biofilms. TcaR is essential for the production of biofilm, which allows bacteria to shield themselves from the immune system and therefore become resistant to antibiotic chemotherapy [21]. Biofilm tolerance is of major clinical significance because most bacterial infections are involved in the formation of biofilm [22]. TcaR is a MarR family protein that can regulate the transcription of *icaADBC* operon, which is important for the production of PIA. Yu-Ming Chang et al. reported in their article that the antibiotics treatment with TcaR in *S. epidermidis* revealed that the biofilm formation was reduced by inhibiting the production of PIA [15]. In addition, they informed that TcaR binds to the DNA sequence in the *icaADBC* operon, and the DNA-binding ability of TcaR is regulated by antibiotics treatment. Therefore, we considered *S. epidermidis* TcaR as a target to design novel drugs that can control bacterial infections, which are a major threat to human life. Herein, by using computer-aided drug design techniques, such as pharmacophore modeling and structure-based drug design, we identified novel inhibitors for *S. epidermidis* TcaR. Pharmacophore modeling is a ligand-based method that can describe the important pharmacophoric features of ligands, which are significant for molecular recognition by a target macromolecule whereas the structure-based lead optimization method involves the generation of pharmacophore models directly from the structure of protein-ligand complex, which is more reliable to understand the required constraints for interaction and specificity. In this study, we utilized the advantage of pharmacophore modeling- (because of its efficiency in virtual screening) and structure-based methodologies to identify the novel inhibitors for *S. epidermidis* TcaR. Predicting the binding affinity of the ligand to a specific target is an important step in the drug design process. Molecular docking is a successful technique to identify the best conformation of a ligand to the specific target [23]. We performed a molecular docking analysis of the selected dataset by utilizing two well-known docking programs: Genetic Optimization for Ligand Docking (GOLD) [24] and AutoDock tools [25]. In continuation, we investigated the pharmacological properties to reveal the drug-likeness of the compounds.

To generate the pharmacophore model, we considered gemifloxacin, an FDA-approved drug, and it is the most efficient drug for bacterial infections compared to all other fluoroquinolone drugs. Especially, it is more effective against methicillin-resistant, methicillin-susceptible *S. epidermidis* species and coagulase-negative staphylococci [26,27]. Before going to start our work, we calculated the binding energies and inhibition constant of methicillin (a known crystallographic inhibitor) and gemifloxacin against *S. epidermidis* TcaR. The gemifloxacin (−10.73 kcal/mol) showed better binding energy than that of methicillin (−6.25 kcal/mol) against *S. epidermidis* TcaR, and the binding mode or orientation of gemifloxacin is different from that of methicillin (Figure 5), which is in good agreement with the literature [26,27]. Binding affinity and binding mode play an important role in rational drug design and development process. Therefore, we considered the best binding affinity drug gemifloxacin for virtual screening of the database via pharmacophore modeling to identify the inhibitors of TcaR. Out of 308 compounds, 16 hits were identified from the molecular docking, and then, these compounds were considered for ADMET analysis to identify the best compounds. Finally, five hits were identified as strong binders of *S. epidermidis* TcaR from the pharmacophore modeling technique.

To perform the structure-based lead optimization, a set of fluoroquinolones that have inhibitory activity against *S. epidermidis* were considered. The pharmacological analysis of the selected dataset revealed that they have some toxic properties, such as irritant, tumorigenic, and low drug-likeness (DL) scores. Hence, we designed new pharmacophore analogs with enhanced DL scores. We further performed a molecular docking analysis of designed compounds on the specific target, *S. epidermidis* TcaR, and identified the best compound, that is, mol34, with good binding energy and DL scores.

The molecular docking analysis of our final hits and gemifloxacin revealed that they extended their interactions with other active site residues, such as ASN20, ALA38, ASN45, VAL63, VAL68, ALA24, VAL43, ILE57, and ARG71, which were not observed in the case of the methicillin-TcaR complex. Whereas, the methicillin interacted with only three residues, namely GLN31, GLN61, and HIS42. Furthermore, our observations on the best two hits, namely ZINC77906236 and ZINC09550296, and their binding modes revealed that the formation of H-bond interactions with ARG110, ASN20, HIS42, and ASN45 active site residues play a promising role in increasing their binding affinity with the active site. Hydrophobic interactions of final hits with hydrophobic residues, such as ALA24, ALA38, VAL63, and VAL68, stabilize the molecules in a particular direction to facilitate a good fit in the active site groove. Our results indicate that the novel identified hit compounds have a strong binding affinity with TcaR and can prevent the binding of the TcaR-DNA complex, leading to reducing the formation of biofilm.

## 4. Materials and Methods

### 4.1. Generation of Pharmacophore Models and Virtual Screening

Pharmacophore modeling is an important factor in rational drug design and the discovery process. ZINCPharmer is an open-source tool that utilizes a new computational approach to pharmacophore search by considering the breadth and complexity of the query compound. Two pharmacophore models of gemifloxacin, which was the most potent drug for bacterial infections compared to all other fluoroquinolone drugs (ciprofloxacin, grepafloxacin, ofloxacin, moxifloxacin, sparfloxacin, and trovafloxacin), against *S. epidermidis* [26,27] were generated by using ZINCPharmer. The technology implemented in ZINCPharmer is a high-performance search engine that uses novel methods, such as geometric hashing, generalized Hough transforms, and Bloom fingerprints, to perform a pharmacophore match that accelerates the search algorithm. To generate the first model, we considered the geometry-optimized structure of gemifloxacin, and the best (lowest binding energy) conformation of gemifloxacin against *S. epidermidis* TcaR was used to generate the second model in common pharmacophore feature search. The pharmacophore feature coordinates and their radii are provided in Appendix A. The large-scale screening in the ZINC15 database matching the 3D-pharmacophore models of gemifloxacin generated 708 hit compounds out of 22 million commercially available purchasable compounds in ready-to-screen formats and with compound vendors’ information. We applied filtration parameters, such as molecular weight less than 500, maximum hits per conformation of each molecule 1, and the number of rotatable bonds < 15, and identified 308 hit compounds, which were satisfied the pharmacophore features of gemifloxacin. ZINCPharmer utilizes the SMARTS matching Open Babel toolkit [28] to identify the possible pharmacophore features, such as H-bond acceptors, H-bond donors, hydrophobic regions, and positive and negative ion charges.

### 4.2. Ligand Preparation and SAR and ADMET Analysis

To perform structure-based lead optimization, we considered a dataset that contains substitution at C-7 position of fluoroquinolone. Substitution or variation on C-7 position of basic fluoroquinolone skeleton increases their antimicrobial activity [22,29] and C-7-substituted piperazine and aminopyrrolidinyl groups leads to development of outstanding fluoroquinolones, such as lomefloxacin and tomefloxacin [22,30]. Therefore, we considered a set of fourteen fluoroquinolones (7a–7p) having antibacterial inhibitory activities [31] (Appendix A) against *Staphylococcus epidermidis* for the present in silico analysis. The molecular structures of the dataset were constructed and energetically minimized by utilizing the molecular mechanics (MM+) force field executed in Hyperchem software (Hypercube 2007, Available online: http://www.hyper.com/ (accessed on 5 February 2022)).

We carried out the pharmacological analysis on the dataset to reveal structural characteristics (H-bond donors/acceptors, lipophilicity, molecular weight, volume, topological polar surface area), physicochemical, and ADMET properties. The molecules satisfying Lipinski’s rule of five [32] have a good bioavailability in the metabolic process of the organism and therefore are more likely to be eligible for oral medications. The pharmacological analysis of these molecules was carried out by using Molinspiration (Available online: http://www.molinspiration.com/ (accessed on 13 March 2022)) and OSIRIS property explorer [33] online tools.

### 4.3. Molecular Docking Simulations

In the molecular docking studies on *Staphylococcus epidermidis* TcaR (PDB ID: 3KP4), we predicted the molecular interaction between ligand and target protein by using well-validated docking programs: GOLD and AutoDock tools. The docking process was applied according to protocols used in our laboratory and described in the previous article [34,35,36]. The ligand-target complex results analysis and interaction images were generated by using Discovery studio visualization software (Discovery Studio visualizer 2012, http://www.accelrys.com/ (accessed on 15 March 2022)). The crystal structure of *Staphylococcus epidermis* TcaR was retrieved from the RCSB Protein databank (Available online: http://www.rcsb.org (accessed on 17 January 2022)) at a resolution of 2.84 Å, and the active site analysis was performed by using SPDBV 4.1.0 Software (Available online: http://www.expasy.org/spdbv/ (accessed on 21 January 2022) [37].

## 5. Conclusions

We performed a comprehensive in silico analysis to design and develop novel inhibitors for *S. epidermidis* TcaR. Pharmacophore modeling-based virtual screening and structure-based lead optimization techniques are very promising methodologies that were used to identify novel inhibitors for specific targets. The FDA-approved drug gemifloxacin was considered to generate the pharmacophore model because it is a potent drug compared to all the other fluoroquinolone drugs and methicillin against *S. epidermidis*. We generated two pharmacophore models of gemifloxacin to virtually screen the ZINC database, and 308 hits were identified. These 308 hits were further screened by using molecular docking simulations and ADMET analysis. Finally, five hit compounds were identified as novel inhibitors of *S. epidermidis* TcaR. To apply the second method, the structure-based drug design, we collected a set of experimentally known inhibitors of *S. epidermidis* from literature to better understand the interaction mechanism of the protein-ligand complex. By applying a series of computational approaches, we identified the hit molecules against *S. epidermis* TcaR for a selected dataset. All the molecules of the dataset fulfilled Lipinski’s rule of five, but their DL scores were very poor. Hence, we carried out the structure-based lead optimization of selected active hits (7a, 7b, and 7g) to design and develop novel pharmacophore analogs with enhanced pharmacological properties, binding affinities, and docking scores. Since the results obtained from our in silico approaches are validated with the experimental inhibitory activity of the selected dataset, this study can reduce efforts of experimentalists in the development of novel antibacterial drugs. We designed a total of 40 molecules and considered them for the in silico analysis. Among 40 molecules designed, five molecules, namely Mol4, Mol9, Mol24, Mol34, and Mol35, showed positive enzyme inhibitor value, good DL scores, and the highest H-bond scores along with good binding affinities. In conclusion, six hit compounds obtained from two methods are considered to be the best antibacterial inhibitors of *S. epidermidis* TcaR. Density functional theory simulations are also conducted to reveal the electronic properties of molecules. Frontier molecular orbital and molecular electrostatic potential surfaces qualitatively explained their crucial role in promoting the inhibitor to adopt a suitable binding orientation in the active site of the target.

## Figures and Tables

**Figure 1 pharmaceuticals-15-00635-f001:**
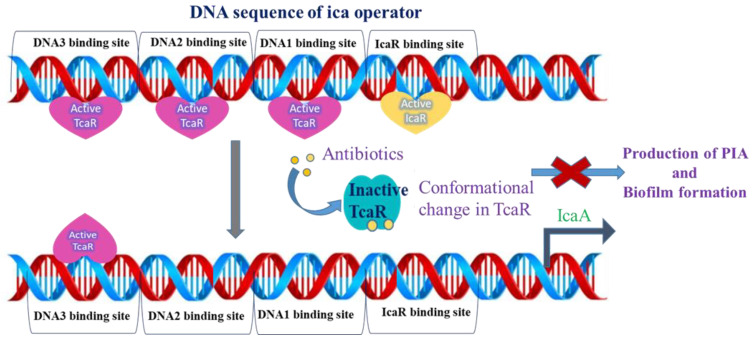
Illustration of antibiotic treatment of *S. epidermidis* TcaR preventing the formation of DNA-TcaR complex in *ica* operon, which leads to inhibition of biofilm formation. The active TcaR can interact with *ica* operator and prevents the transcription of *IcaA*. Upon treating with antibiotics, significant conformational changes in the DNA binding domain of TcaR will occur.

**Figure 2 pharmaceuticals-15-00635-f002:**
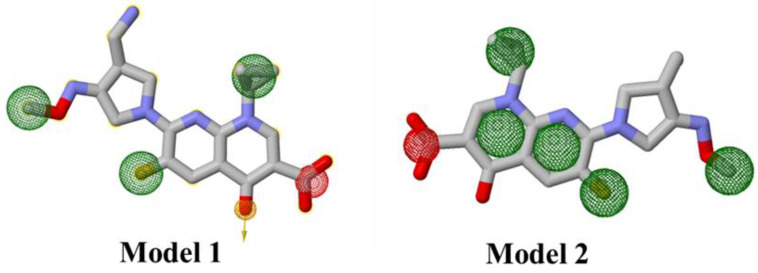
Representation of two gemifloxacin pharmacophore models. The green, red, and yellow color interlaces indicate the hydrophobic, negative ion charge, and hydrogen bond acceptor regions, respectively.

**Figure 3 pharmaceuticals-15-00635-f003:**
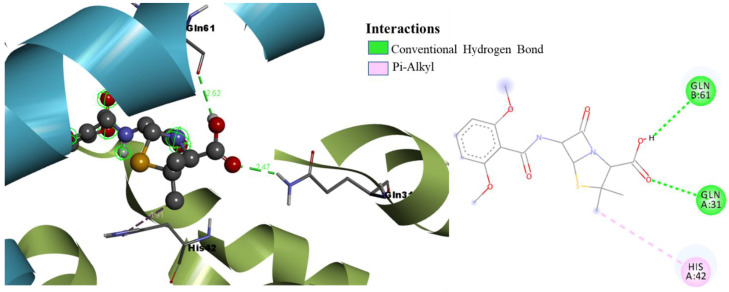
Binding mode conformation and molecular interactions of methicillin in the active site of *S. epidermidis* TcaR. The left and right figures represent the molecular interactions in 3D and 2D, respectively. The methicillin is shown in the ball-stick model, and the key interacting residues are shown as grey sticks.

**Figure 4 pharmaceuticals-15-00635-f004:**
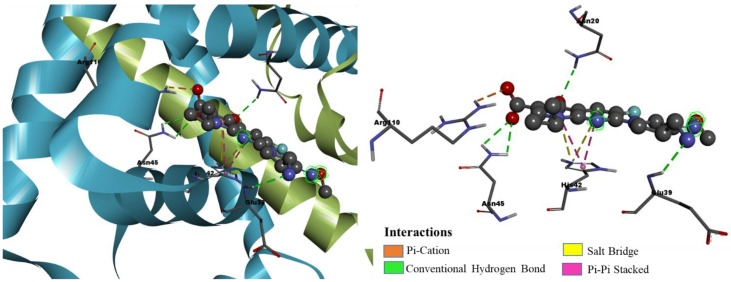
Binding mode conformation and molecular interactions of gemifloxacin in the active site of *S. epidermidis* TcaR. The gemifloxacin is shown in the ball-stick model, and the key interacting residues are shown as grey sticks.

**Figure 5 pharmaceuticals-15-00635-f005:**
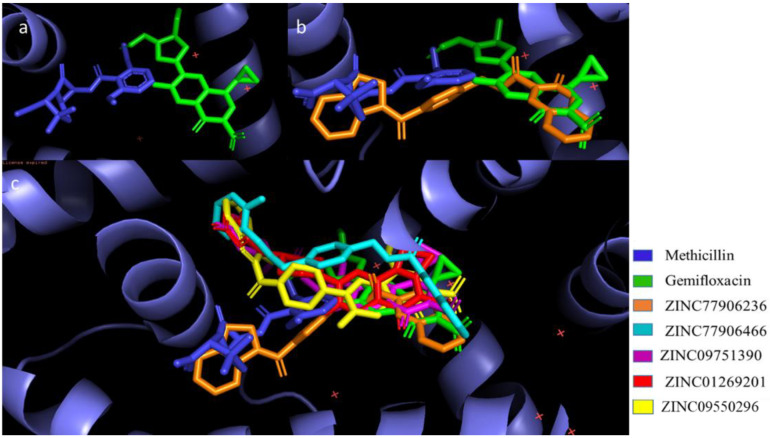
Diagram of the binding mode of methicillin and gemifloxacin (**a**). Superimposition of hit compound ZINC77906236 on methicillin and gemifloxacin in the active site of TcaR (**b**). The binding mode of methicillin, gemifloxacin, and hit compound ZINC77906236 (**c**). The binding mode of methicillin, gemifloxacin, and hit compounds (ZINC77906236, ZINC09550296, ZINC77906466, ZINC09751390, and ZINC01269201) in the active site of TcaR.

**Figure 6 pharmaceuticals-15-00635-f006:**
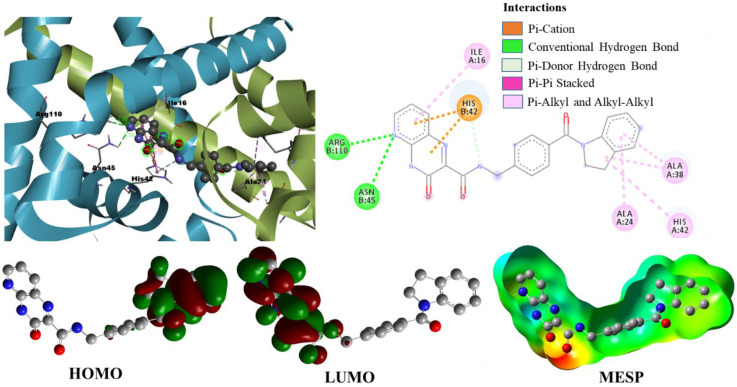
Binding mode conformation and molecular interactions of ZINC77906236 in the active site of *S. epidermidis* TcaR. The left and right figures represent the molecular interactions in 3D and 2D, respectively. The hit compound is shown in the ball-stick model, and the key interacting residues are shown as grey sticks. Isodensity plots of HOMO and LUMO and molecular electrostatic potential of hit compound.

**Figure 7 pharmaceuticals-15-00635-f007:**
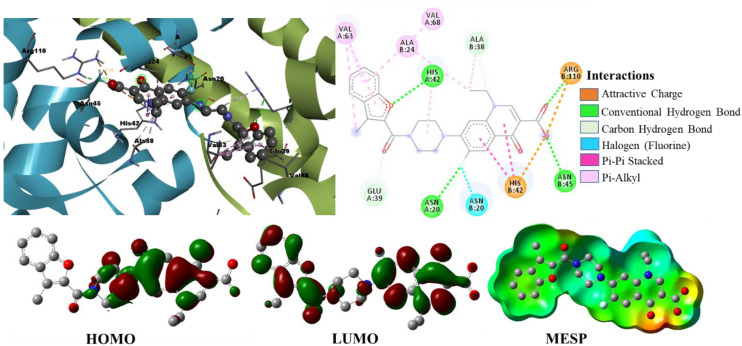
Binding mode conformation and molecular interactions of hit compound ZINC09550296 in the active site of *S. epidermidis* TcaR. The left and right figures represent the molecular interactions in 3D and 2D, respectively. The hit compound is shown in the ball-stick model, and the key interacting residues are shown as grey sticks.

**Figure 8 pharmaceuticals-15-00635-f008:**
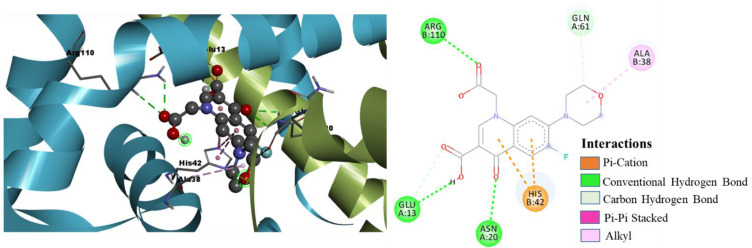
Binding mode conformation and molecular interactions of compound 7a in the active site of *S. epidermidis* TcaR. The left and right figures represent the molecular interactions in 3D and 2D, respectively. The compound 7a is shown in the ball-stick model, and the key interacting residues are shown as grey sticks.

**Figure 9 pharmaceuticals-15-00635-f009:**
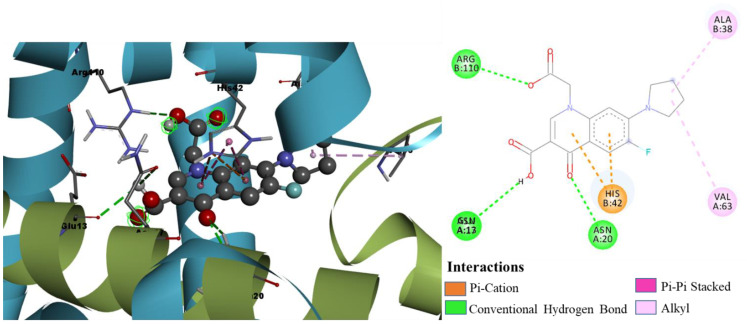
Binding mode conformation and molecular interactions of compound 7b in the active site of *S. epidermidis* TcaR. The left and right figures represent the molecular interactions in 3D and 2D, respectively. The compound 7b is shown in the ball-stick model, and the key interacting residues are shown as grey sticks.

**Figure 10 pharmaceuticals-15-00635-f010:**
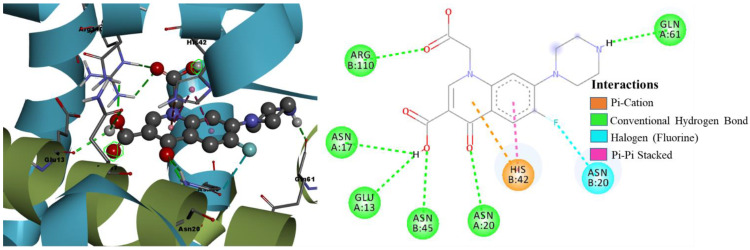
Binding mode conformation and molecular interactions of compound 7g in the active site of *S. epidermidis* TcaR. The left and right figures represent the molecular interactions in 3D and 2D, respectively. The compound 7g is shown in the ball-stick model, and the key interacting residues are shown as grey sticks.

**Figure 11 pharmaceuticals-15-00635-f011:**
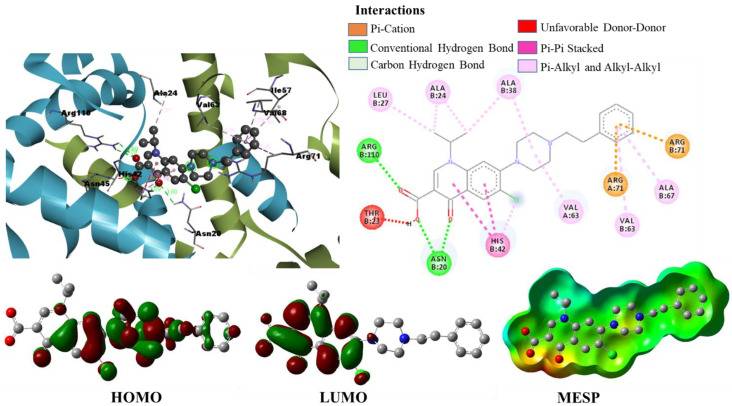
Binding mode conformation and molecular interactions of compound Mol34 in the active site of *S. epidermidis* TcaR. The left and right figures represent the molecular interactions in 3D and 2D, respectively. The hit compound is shown in the ball-stick model, and the key interacting residues are shown as grey sticks. Isodensity plots of HOMO and LUMO and molecular electrostatic potential of hit compound.

**Table 1 pharmaceuticals-15-00635-t001:** Binding energies and estimated inhibition constants of hit compounds identified from pharmacophore-based virtual screening. Methicillin is a known crystallographic inhibitor of *S. epidermidis* TcaR.

S. No	Compounds	Binding Energy (kcal/mol)	Estimated Inhibition Constant (Ki)
1	ZINC77906236	−13.27	187.61 pM
2	ZINC03114214	−13.07	260.55 pM
3	ZINC09550296	−12.89	353.69 pM
4	ZINC77906466	−12.74	460.70 pM
5	ZINC01958447	−12.68	507.91 pM
6	ZINC09751390	−12.38	843.81 pM
7	ZINC01269201	−12.34	895.54 pM
8	ZINC21985520	−12.13	1.29 nM
9	ZINC09751395	−12.08	1.39 nM
10	ZINC02280291	−12.07	1.41 nM
11	ZINC01440193	−11.93	1.79 nM
12	ZINC01127091	−11.91	1.85 nM
13	ZINC72332562	−11.57	3.31 nM
14	ZINC00794058	−11.56	3.37 nM
15	ZINC00686337	−11.40	4.40 nM
16	ZINC09550295	−11.36	4.71 nM
17	Gemifloxacin	−10.73	13.72 nM
18	Methicillin	−6.25	26.35 uM

**Table 2 pharmaceuticals-15-00635-t002:** Molecular docking results of selected dataset against *S. epidermidis* TcaR.

Molecule	Binding Energy (kcal/mol)	Fitness Score	S(hb_ext) ^a^	S(vdw_ext) ^b^	S(vdw_int) ^c^
7a	−8.7	56.33	6.27	45.78	−12.89
7b	−9.0	57.47	6.73	44.81	−13.19
7c	−8.9	61.30	2.01	49.34	−8.53
7d	−9.3	59.35	3.52	50.67	−13.84
7e	−8.3	60.04	6.09	52.25	−12.65
7f	−9.6	58.97	1.94	51.08	−13.22
7g	−9.2	56.90	6.40	47.19	−14.39
7h	−8.7	58.52	3.01	50.26	−22.09
7i	−8.8	61.34	2.70	54.17	−15.85
7j	−9.7	61.63	4.68	52.34	−15.01
7k	−8.5	62.09	2.26	55.70	−16.26
7l	−7.6	60.19	4.72	51.96	−15.98
7n	−4.3	58.06	4.76	47.51	−12.02
7o	−9.2	59.58	5.97	51.99	−17.88
7p	−9.2	57.13	6.04	51.89	−16.03

^a^ Protein-ligand H-bond scores; ^b^ Protein-ligand van der Waals scores; ^c^ Intramolecular van der Waals strain within the ligand.

**Table 3 pharmaceuticals-15-00635-t003:** The intermolecular interactions of hit compounds and gemifloxacin in the active site of *S. epidermidis* TcaR. The corresponding bond lengths (Å) are shown in parenthesis.

Compound	H-Bond Interactions	Hydrophobic and Other Interactions
ZINC77906236	ARG110 (2.86, 2.76), ASN45 (2.79)	ILE16 (5.40), ALA24 (4.91, 4.68), ALA38 (4.09, 4.85), HIS42 (3.89, 4.27, 4.43, 3.97)
ZINC09550296	ASN20 (2.57), HIS42 (2.79), ASN45 (2.56), ARG110 (2.55)	ALA24 (4.05), ALA38 (3.67, 3.67), GLU39 (3.50), HIS42 (4.65, 4.11, 4.16), VAL63 (4.68, 5.16, 4.57), VAL68 (5.12), ARG110 (3.15)
ZINC77906466	ALA24 (2.96), ASN45 (2.70)	ASN20 (3.72), THR23 (3.87), HIS42 (2.93), VAL43 (5.37), ILE57 (5.37), VAL63 (5.05), VAL68 (5.0), ARG71 (5.25), ARG110 (3.59, 4.11)
ZINC09751390	ASN20 (2.29), THR23 (2.38), HIS42 (2.62), ASN45 (2.68), ARG71 (2.69)	ALA38 (3.73, 4.65), HIS42 (3.83, 4.07, 4.30, 4.91), VAL63 (4.85), ARG110 (2.97)
ZINC01269201	THR23 (2.91), HIS42 (2.24), ASN45 (2.72)	ALA38 (3.44), HIS42 (4.60, 4.40, 4.54, 4.74), VAL63 (3.72, 5.42), VAL68 (4.58), ARG110 (3.23)
Mol34	ASN20 (3.00), HIS42 (3.09), ARG110 (2.69, 2.79)	ALA24 (4.31), HIS42 (3.90, 4.18, 3.09), ILE57 (5.48), VAL63 (4.85), VAL68 (4.89), ARG71 (5.20)
Gemifloxacin	ASN20 (2.54), GLU39 (2.76), ASN45 (2.69, 2.67)	HIS42 (3.65, 4.01, 3.98, 4.09), ARG110 (2.60)
Methicillin	GLN31 (2.47), GLN61 (2.62)	HIS42 (4.91)

## Data Availability

Data are contained within the article or Appendix A.

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
