# Peer review of "A Combination of Pharmacophore-Based Virtual Screening, Structure-Based Lead Optimization, and DFT Study for the Identification of S. epidermidis TcaR Inhibitors"

_pharmaceuticals, 2022, doi:10.3390/ph15050635_

Round 1

Reviewer 1 Report

The authors in this manuscript exerted a respected effort to develop and optimize new anti-biofilm agents. The authors based all their findings on the in-silico results, from our extended deep studies in this field, it is a must to prove the in-silico results and support them with in-vitro and in-vivo experiments. Despite my conviction of the quality of the presented work, it lakes the essential proof.  

Author Response

Point: The authors in this manuscript exerted a respected effort to develop and optimize new anti-biofilm agents. The authors based all their findings on the in-silico results, from our extended deep studies in this field, it is a must to prove the in-silico results and support them with in-vitro and in-vivo experiments. Despite my conviction of the quality of the presented work, it lakes the essential proof.

Response: We have designed and developed novel anti-biofilm agents based on in-silico simulations. As we work only on computational simulations and due to the lack of in-vitro and/or in-vivo experimental facilities, we cannot carry out any experimental studies. However, our selected dataset (7a – 7p) of 14 molecules are experimentally proven antibiotics for the selected target, S. epidermis. We have also validated our in silico methodology and results with the experimental inhibitory activity. Also, we performed the virtual screening of the ZINC database where we have used an efficient and FDA-approved drug, Gemifloxacin as a pharmacophore. The ZINC database contains biologically active conformations. We strongly believe that the present study is suitable for this journal and in particular for the special issue “Drug Candidates for the Treatment of Infectious Diseases”.

Reviewer 2 Report

The paper is overall well designed and presented. the methods used are reproducible and easy to follow. still some points need to addressed below.

- Please consider adding line numbers for easy referring in the text and also reformat the paper following the journal template

- Abstract: Please consider deleting unnecessary information to give some space for the results. No results description in your abstract ( active site residue, compounds with the highest score)

- Figure 1 idea is good. However, it is a bit confusing. As you showed an active and inactive TcaR, you should describe the outcome of each one. The active one will result in the formation of PIA and biofilm, while the inactivation inhibits the process. Enhance the resolution of the figure in the meantime

In general, the introduction contextualizes well the study, and the aims are clear

2.1.1.

- Can you add here the details of the ‪filtering parameters

- Figure S1 is a bit confusing. PDB ID 3KP4 is the Cristal structure of TcaR with methicillin. And you tried to test your g‪rid ‪dimensions with the tested methicillin structure. The approach is perfect. However, the figure needs some color adjustment or maybe adding some descriptive arrow so we can easily differentiate the content.

- How was the ‪Estimated inhibition constant calculated

- “‪Finally, from a series of ‪computational simulations, we have identified a total of five compounds which include ‪ZINC77906466 and ZINC77906236 (non-fluoroquinolones), ZINC09751390, ZINC09550296, ‪and ZINC01269201 (prulifloxacin” on what basis?

- I was hoping to see a comparison of the binding mode of the 3 first compounds with ‪methicillin and gemifloxacin to have a view of what caused that considerable difference in Binding Energy

- Gemifloxacin binding energy in table 1 is -6.66 kcal/mol, While in the text you say that it is ‪-10.73 kcal/mol . which one is correct‪?

2.1.2.3.

‪- “ Detailed ‪investigation of the molecular interactions revealed that gemifloxacin (Figure S2) formed four H-bonds and one π-cation interaction with ASN45, ASN20, GLU39, and ARG110, respectively.”

In figure S2, it is clear that there is only 3 H-bonds ASN45, ASN20 and GLU39. Also, from the same figure, it is not clear where the salt bridge and Pi-cation are, as they are both with the same colors. This figure is important. Please move it to the main text and consider adding one for methicillin

- I was looking for the ‪Binding mode analysis of methicillin too to compare the bonds and interactions with the other compounds. Please consider adding it and comparing it with at least gemifloxacin binding mode analysis in the text

‪2.1.2.3.1. Binding mode analysis of hit compound ZINC77906236

‪- “this compound has formed two H-bond ‪interactions with ARG110 and one H-bond interaction with ASN45.” Do you mean this compound has formed two H-bond ‪interactions with ARG110 and ASN45 ?

- I’m a bit confused how for example compound ZINC77906236 has the highest Binding energy (-13.27 kcal/mol) with less H-bonds (2) formed compared to gemifloxacin (3) and also ZINC09550296 (4). Also what is the significance of the obtained Ki. 187.61 pM vs 13.03 uM for example.

- “Based on our observations and ‪docking results, this hit (ZINC77906236) can act as a competitive inhibitor of both drugs.” In the context of your study even if that maybe correct, defining a competitive and a non-competitive inhibitor requires some practical investigation rather than a docking simulation.

- “nowadays fluoroquinolone antibacterial drug ‪resistance becomes a major issue” reference required

2.2.2.

- Please make a good introduction on how were the molecules 7a-p selected and refer to S8 for their structure. It is bit confusing when you jump to it right away.

- In this part of the methods “We took a set of fourteen fluoroquinolones having antibacterial inhibitory activities [29](Table S8) against Staphylococcus epidermidis for the present in silico analysis.” Adding “fourteen fluoroquinolone (7a-7p)” will be better.

- The molecule 7j has the highest Binding Energy  ‪Fitness score S(vdw_ext) S(vdw_int) and an average S(hb_ext). Why was it considered an inactive molecule ? Can you please provide the Binding mode conformation and molecular interactions of it?

- DL, define when it first appears in the text.

2.2.3.

- Why precisely positions 1, 6, and 7?

- Table 4. Replace ZINC22059926 with gemifloxacin and add methicillin data too

Author Response

Point 1: Please consider adding line numbers for easy referring in the text and also reformat the paper following the journal template.

Response 1: We have formatted the paper into the template of the journal.

Point 2: Abstract: Please consider deleting unnecessary information to give some space for the results. No results description in your abstract (active site residue, compounds with the highest score)

Response 2: We have deleted unnecessary information from the abstract and included the results, active site residues information and the compounds with the highest score.

Point 3:  Figure 1 idea is good. However, it is a bit confusing. As you showed an active and inactive TcaR, you should describe the outcome of each one. The active one will result in the formation of PIA and biofilm, while the inactivation inhibits the process. Enhance the resolution of the figure in the meantime.

Response 3: We have modified Figure 1 according to the reviewer’s suggestion.

Point 4:  Can you add here the details of the ‪filtering parameters

Response 4: We have included the details of filtering parameters on page 4 (lines 119-121) of the revised manuscript.

Point 5:  Figure S1 is a bit confusing. PDB ID 3KP4 is the Cristal structure of TcaR with methicillin. And you tried to test your grid ‪dimensions with the tested methicillin structure. The approach is perfect. However, the figure needs some color adjustment or maybe adding some descriptive arrow so we can easily differentiate the content.

Response 5: We have described the colors in the caption of Figure S1.

Point 6:  How was the ‪Estimated inhibition constant calculated

Response 6:We have described the details of inhibition constant calculation on page 4 (138 - 142) of the revised manuscript.

Point 7:  “‪Finally, from a series of ‪computational simulations, we have identified a total of five compounds which include ‪ZINC77906466 and ZINC77906236 (non-fluoroquinolones), ZINC09751390, ZINC09550296, ‪and ZINC01269201 (prulifloxacin)” on what basis?

Response 7: These five compounds were identified based on Lipinski’s rule of five, drug-like score, and binding energy. We have included the criterion for identification on page 5 (lines 171 – 173) in the revised manuscript.

Point 8: I was hoping to see a comparison of the binding mode of the 3 first compounds with ‪methicillin and gemifloxacin to have a view of what caused that considerable difference in Binding Energy.

Response 8: The last paragraph of the discussions section, on page 16 (lines 470 – 482), discusses the comparison of top five hits with gemifloxacin and methicillin in terms of interacting residues. We have explained how the extended interactions of the top five hits with the active residues facilitated improving binding energy.

Point 9: Gemifloxacin binding energy in table 1 is -6.66 kcal/mol, While in the text you say that it is ‪-10.73 kcal/mol. which one is correct‪?

Response 9: The correct value is -10.73 kcal/mol. We have corrected the value in Table 1.

Point 10:  In figure S2, it is clear that there is only 3 H-bonds ASN45, ASN20 and GLU39. Also, from the same figure, it is not clear where the salt bridge and Pi-cation are, as they are both with the same colors. This figure is important. Please move it to the main text and consider adding one for methicillin. I was looking for the ‪Binding mode analysis of methicillin too to compare the bonds and interactions with the other compounds. Please consider adding it and comparing it with at least gemifloxacin binding mode analysis in the text

Response 10: As per the reviewer's suggestion, we have moved Figure S2 to the main manuscript. Also, we have made changes to the salt bridge color. In the modified figure the 4 H-bonds (with ASN45, ASN20, GLU39, and ARG110) can be seen.  In addition, we have added the methicillin binding mode in the main manuscript.

Point 11: ‪2.1.2.3.1. Binding mode analysis of hit compound ZINC77906236. - “this compound has formed two H-bond ‪interactions with ARG110 and one H-bond interaction with ASN45.” Do you mean this compound has formed two H-bond ‪interactions with ARG110 and ASN45?

Response 11: We would like to clarify this comment. Here, two H-bonds formed with ARG110 and another H-bond formed with ASN45. In Figure 6, the two H-bonds (with ARG 110) are overlapped in 2D representation but can be seen clearly in 3D representation.

Point 12: I’m a bit confused how for example compound ZINC77906236 has the highest Binding energy (-13.27 kcal/mol) with less H-bonds (2) formed compared to gemifloxacin (3) and also ZINC09550296 (4). Also what is the significance of the obtained Ki. 187.61 pM vs 13.03 uM for example.

Response 12: The compound ZINC77906236 has formed three H-bonds (two with ARG110 and one with ASN45). The binding energy does not only depend on H-bonds but also several other factors such as intermolecular energy (vdW + H-bond + desolvation + electrostatic), total internal energy, and torsional free energy also contribute significantly. The obtained Ki of 187.61 pM is better than 13.03 μM.

Point 13: “Based on our observations and ‪docking results, this hit (ZINC77906236) can act as a competitive inhibitor of both drugs.” In the context of your study even if that maybe correct, defining a competitive and a non-competitive inhibitor requires some practical investigation rather than a docking simulation.

Response 13: We have removed the sentence in the revised manuscript.

Point 14: “nowadays fluoroquinolone antibacterial drug ‪resistance becomes a major issue” reference required

Response 14: Two references (17 and 18) are added in the revised manuscript.

Point 15:  Please make a good introduction on how were the molecules 7a-p selected and refer to S8 for their structure. It is bit confusing when you jump to it right away.- In this part of the methods “We took a set of fourteen fluoroquinolones having antibacterial inhibitory activities [29] (Table S8) against Staphylococcus epidermidis for the present in silico analysis.” Adding “fourteen fluoroquinolone (7a-7p)” will be better.

Response 15: We made an introduction on how the molecules were considered on pages 14 and 15 (lines 481 – 487) in the revised manuscript. The additional changes were also made as per the reviewer's suggestion.

Point 16: The molecule 7j has the highest Binding Energy  ‪Fitness score S(vdw_ext) S(vdw_int) and an average S(hb_ext). Why was it considered an inactive molecule? Can you please provide the Binding mode conformation and molecular interactions of it?

Response 16: Based on our observation, the active molecules (7a, 7b, 7g, and Mol34) interact with the active site residues ARG110, ASN20, ASN45, and HIS42 and formed strong H-bond interactions. On the other hand, the molecule 7j has not formed any H-bond interactions with the above-mentioned residues but formed two H-bond interactions with ALA24 (2.79 Å) and THR23 (2.16 Å). Despite having high binding energy, the 7j molecule has not formed specific interactions which were observed in the case of 7a, 7b, 7g, and Mol34. The binding mode conformation and its molecular interactions are given in the supporting information (Figure S4).

Point 17:  DL, define when it first appears in the text.

Response 17: We have defined the DL on its first appearance.

Point 18: Why precisely positions 1, 6, and 7?

Response 18: We have explained and added three references on page 11 (lines 313 - 317) in the revised manuscript.

Point 19: Table 4. Replace ZINC22059926 with gemifloxacin and add methicillin data too.

Response 19: We have modified Table 4 accordingly.

Reviewer 3 Report

The article received for evaluation is titled 'A Combination of Pharmacophore...' and has as main topic the structure-based drug design and DFT simulation in order to find best candidates for S. epidermis and S. aureus  novel inhibitors.

First observation is that the manuscript is not arranged into the requested template, that sometimes make it difficult to follow or to have a glimpse about the whole work. Abstract is a bit too long containing unnecessary data. The Introduction is well written and contains 16 relevant References.

As authors used gemifloxacin, considering it the most potent drug for S. epidermidis bacterial infections, literature data should be added to support this statement. Also, a virtual comparison with other drugs used for similar purposes should be present.

More details about ZINC and AutoDock should be added, these being required for reproducibility by other scientists. However, very good quality pictures are noticed throughout the manuscript. I also suggest to the authors to number the subchapter about Density Functional Theory calculations.

Regarding the software, Molinspiration is still actual, but Hyperchem ceased to be updated long time ago. However, its supply accurate data. 35 references ends the papaer, DOI number should be added as well. A supplementary materials is also presented, including 16 pages.

As a conclusion, the work is well done and well presented, subject to minor changer, therefore should be accepted for publication.

Author Response

Point 1:  First observation is that the manuscript is not arranged into the requested template, that sometimes make it difficult to follow or to have a glimpse about the whole work. Abstract is a bit too long containing unnecessary data. The Introduction is well written and contains 16 relevant References.

Response 1: We have formatted the manuscript into the journal format. The abstract is modified by removing unnecessary data.

Point 2: As authors used gemifloxacin, considering it the most potent drug for S. epidermidis bacterial infections, literature data should be added to support this statement. Also, a virtual comparison with other drugs used for similar purposes should be present.

Response 2:  As per the reviewer’s suggestion, we have added the relevant references (26 and 27) in which the gemifloxacin was reported as the potent drug over ciprofloxacin, grepafloxacin, ofloxacin, moxifloxacin, sparfloxacin, and trovafloxacin against S. epidermidis.

Point 3: More details about ZINC and AutoDock should be added, these being required for reproducibility by other scientists. However, very good quality pictures are noticed throughout the manuscript. I also suggest to the authors to number the subchapter about Density Functional Theory calculations.

Response 3:  More details of ZINC and AutoDock were added in the revised manuscript. Table S8 of the supporting material summarizes the pharmacophore feature coordinates and their radius. The Density Functional Theory calculations section has been numbered as 2.3 in the revised manuscript.

Point 4: Regarding the software, Molinspiration is still actual, but Hyperchem ceased to be updated long time ago. However, its supply accurate data. 35 references ends the papaer, DOI number should be added as well. A supplementary materials is also presented, including 16 pages.

As a conclusion, the work is well done and well presented, subject to minor changer, therefore should be accepted for publication.

Response 4: As per the reviewer’s suggestion we have added the DOI in the references section. The total number of references is now 37 in the revised manuscript.

Round 2

Reviewer 1 Report

I believe that the significance of the study, however, the biological proof is necessary. Yes, the compounds 7a-7p are proven by others, but what is the proof of your modification.

Reviewer 2 Report

The revisions were done correctly.

One last thing, please mention in the footnote if a bond is overlapped in the 2D scheme so it become clearer.